

# Registered report: the androgen receptor induces a distinct transcriptional program in castration-resistant prostate cancer in man

Denise Chronscinski[1], Srujana Cherukeri[1], Fraser Tan[2], Nicole Perfito[2], Joelle Lomax[2] and Elizabeth Iorns[2]

[1] Noble Life Sciences, Inc., Gaithersburg, MD, USA
[2] Science Exchange and The Prostate Cancer Foundation-Movember Foundation Reproducibility Initiative, Palo Alto, CA, USA

## ABSTRACT

The Prostate Cancer Foundation-Movember Foundation Reproducibility Initiative (PCFMFRI) seeks to address growing concerns about reproducibility in scientific research by conducting replications of recent papers in the field of prostate cancer. This Registered Report describes the proposed replication plan of key experiments from "The Androgen Receptor Induces a Distinct Transcriptional Program in Castration-Resistant Prostate Cancer in Man" by Sharma and colleagues (*2013*), published in *Cancer Cell* in 2013. Of thousands of targets for the androgen receptor (AR), the authors elucidated a subset of 16 core genes that were consistently down-regulated with castration and re-emerged with castration resistance. These 16 AR binding sites were distinct from those observed in cells in culture. The authors suggested that cellular context can have dramatic effects on downstream transcriptional regulation of AR binding sites. The present study will attempt to replicate Fig. 7C by comparing gene expression of the 16 core genes identified by Sharma and colleagues in xenograft tumor tissue compared to androgen treated LNCaP cells *in vitro*. The Prostate Cancer Foundation-Movember Foundation Reproducibility Initiative is a collaboration between the Prostate Cancer Foundation, the Movember Initiative, and Science Exchange, and the results of the replications will be published by *PeerJ*.

## INTRODUCTION

In tissues where it is expressed, the ligand-activated androgen receptor (AR) binds to DNA response elements and directs expression of cell- and tissue- specific genes (*Kang, Janne & Palvimo, 2004*; *Wang, Carroll & Brown, 2005*). In order to form a functional transcription complex, AR employs co-regulator proteins and transcription factors to direct binding and influence transcription (*Heinlein & Chang, 2002*). Prostate cancer (PC) cells depend on steroid hormones and AR to mediate oncogenic growth (*Wang et al., 2009*), and therefore, one of the key treatments for prostate cancer following surgery and radiation therapy is androgen deprivation therapy (ADT). ADT greatly reduces levels of the endogenous ligand

Corresponding author
Nicole Perfito,
nicole@scienceexchange.com

for the androgen receptor and thus affects its target genes, hindering their ability to drive oncogenic growth (reviewed in *Lian et al., 2015*).

While initially effective, the disease often returns despite ADT; it is then termed castration-resistant prostate cancer (CRPC), an aggressive and fatal form of prostate cancer. The mechanisms of CRPC are not completely understood, but activation of the AR may play an important role in the transition to CRPC (*Lin, Gout & Wang, 2013*). Whole genome analyses point to genomic rearrangements and alterations in the transcriptional program particularly related to AR signaling in CRPC cells (*Shtivelman, Beer & Evans, 2014*). Functional outcomes of these alterations include increased AR mRNA and protein in CRPC cells (*Yuan et al., 2014*), the ability for other hormones to bind AR (*Taplin et al., 1999*), generation of ligand independent constitutively active AR variants (*Ware et al., 2014*; *Ferraldeschi et al., 2015*), and increased sensitivity of AR by posttranslational modifications (*Guo et al., 2006*).

In their 2013 *Cancer Cell* paper, Sharma and colleagues (*2013*) used chromatin immunoprecipitation sequencing (ChIP-seq) to identify genome-wide binding targets of AR in prostate tissue from patients with CRPC, ADT-responsive PC, or untreated PC, compared with benign prostate hyperplasia. This analysis identified thousands of AR target genes. Of the 150 genes that were upregulated in untreated PC compared to CRPC, and downregulated with castration in PC xenografts, 16 core genes consistently re-emerged with castration resistance: *PECI*; *TNFSF10*; *ABHD12*; *XRCC3*; *MAD1L1*; *SEC61A1*; *GFM1*; *TSPAN13*; *STIL*; *TRMT12*; *EIF2B5*; *TM4SF1*; *NDUFB11*; *SLC26A2*; *AGR2*; *NT5DC3*. CRPC is often studied *in vitro* using cell lines derived from metastatic cancer cells (*Wang et al., 2009*; *Yu et al., 2010*; *Massie et al., 2011*), and previous authors have suggested PC cells in culture have different gene expression signatures than primary tumors (*Yu et al., 2009*; *Long et al., 2014*). Sharma and colleagues showed that genome-wide binding targets of the AR in tumor tissue differed from those in cultured cells. Expression patterns of the 16 core AR target genes were distinct from those observed in cells in culture, suggesting cellular context can have dramatic effects on downstream transcriptional regulation of AR binding sites.

In Fig. 7C, expression levels of the 16 core genes were analyzed in xenograft tumor tissues that were determined to be either castration-sensitive or castration-resistant. Gene expression signatures in these tumors were compared to tumors from intact mice. The core genes were downregulated in castration-sensitive tumors compared to tumors from intact animals, and upregulated in castration-resistant tumors (*Sharma et al., 2013*). This is a key finding and will be replicated in Protocols 2 and 3. Another key finding of this study was that the expression patterns of the core genes in tumor tissues were distinct from their expression pattern in LNCaP cell culture. In order to replicate this key finding, we will also compare expression patterns of the core gene set in both castration-sensitive and castration-resistant tumors to the genetic signature of these genes in cultured LNCaP cells with or without androgen exposure *in vitro* in Protocols 1 and 3.

A large number of studies have investigated genome-wide profiling of AR binding (*Bolton et al., 2007*; *Jia et al., 2008*; *Wang et al., 2009*; *Yu et al., 2010*). The study by Sharma and colleagues is to some extent a re-analysis of AR regulation in PC xenografts

from data originally presented by Terada and colleagues (*2010*). Tao and colleagues (*2014*) also profiled AR target genes in cultured LNCaP cells but reported gene ontology only. Some of the 16 core AR-binding targets identified in *Sharma et al. (2013)* have been studied as markers for prostate cancer, either by mutation (*XRCC3*; *Xuan, Hui & Fang, 2015*) or increased splice variant expression (*AGR2*; *Neeb et al., 2014*). Some of the core 16 targets have been identified as targets for AR in breast cancer (*MAD1L1*; *Mehta et al., 2015*), in association with radiation-resistant lung adenocarcinoma cells (*TM4SF5*; *Choi et al., 2014*) and pancreatic ductal adenocarcinoma (*AGR2*; *Mizuuchi et al., 2015*). Lastly, an antibody targeting TM4SF1 reduced human prostate cancer cells in matrigel implants *in vivo* (*Lin et al., 2014*).

# MATERIALS & METHODS

An asterisk (*) indicates data or information provided by the PCFMFRI core team. A hashtag (#) indicates information provided by the replicating lab. All other protocol information was derived from the original paper, references from the original paper, or from communication from the original authors. All references to Figures are in reference to the original study.

## Protocol 1: Stimulation of cultured LNCaP cells with androgen treatment

This protocol describes how to stimulate cultured LNCaP human prostate cancer cells with androgen in order to assess the expression level of the core set of 16 genes identified by Sharma and colleagues (*2013*). This is an additional experiment added by the PCFMFRI core team designed to probe for differences in gene signature between cultured cells and tumors *in vivo*, and is based upon work performed by Massie and colleagues (*2011*). Cells produced in this protocol will be used for gene expression analysis in Protocol 3.

### *Sampling*

- Each experiment consists of two cohorts:
  - Cohort 1: untreated LNCaP cells
  - Cohort 2: LNCaP cells treated with R1881 (synthetic androgen)
- The experiment will be performed 3 times
  - This experiment is exploratory in nature, and thus no power calculations are necessary.

### Materials and reagents

| Reagent | Type | Manufacturer | Catalog # | Comments |
|---|---|---|---|---|
| LNCaP cells, clone FGC | Cells | ATCC | CRL-1740 | Original unspecified |
| RPMI-1640 (Phenol-red free) | Media | Life Technologies | 11835-030 | Replaces ATCC 30-2011 |
| Fetal bovine serum (charcoal dextran-stripped) | Reagent | Sigma-Aldrich | F4135 | Original source unspecified |
| 6 cm tissue culture dishes | Labware | Will be left to the discretion of the replicating lab | | Original source unspecified |
| R1881 | Synthetic androgen | Sigma-Aldrich | R0908 | |
| DMSO | Reagent | Sigma-Aldrich | D8418 | Original source unspecified |
| TRIzol® Reagent | Reagent | Life Technologies | 15596 | Original not specified |

### Procedure

Notes:

- All cells will be sent for mycoplasma testing and STR profiling
1. Plate $1 \times 10^5$ LNCaP cells in 6 cm tissue culture dishes and let grow overnight to adhere.
    (a) LNCaP cells are expanded in RPMI-1640 supplemented with 10% FBS at 37 °C/5% $CO_2$. When cells have reached approximately 70% confluence, media will be removed, cells washed twice with 1x PBS and replaced with RPMI media supplemented with 10% charcoal dextran treated FBS.
    (b) Prepare 2 separate plates; one for treatment, one for control.
2. Cells are kept in steroid depleted media for 72 h prior to androgen stimulation or vehicle control treatment. Treat cells with R1881 or DMSO for 0 and 24 h.
    (a) For treatment, replace media with fresh media containing 1 nM R1881.
        i. R1881 is dissolved in DMSO at >10 mg/ml.
    (b) For control, add vehicle (DMSO) only, at the same total volume as for treated cells.
3. [#] Lyse cells at 0 and 24 h time points with TRIzol reagent, according to manufacturer's instructions.
    (a) Use mechanical disruption (i.e., cell scraping) to dislodge cells, if necessary.
4. #Immediately flash-freeze and store cell/TRIzol mixture at −80 °C until ready to perform RNA extraction in Protocol 3.
5. Repeat experiment an additional two independent times.

### Deliverables

- Data to be collected:
    ○ None applicable
- Samples delivered for further analysis:
    ○ Homogenized cells for use in Protocol 3, stored at −80 °C

### Confirmatory analysis plan

- See Protocol 3 for analysis of gene expression

### Known differences from the original study

- This experiment was not performed in the original study; it is an additional experiment added by the PCFMFRI core team in order to explore a the key finding that cultured cells possess a different AR-mediated gene signature for the core 16-gene set than *in situ* tumor tissue (*Sharma et al., 2013*). The FGC clone of the LNCaP cell line was not used in the study by Tao and colleagues, but rather low passage LNCaP cells.

### Provisions for quality control

All data obtained from the experiment—raw data, data analysis, control data and quality control data—will be made publicly available, either in the published manuscript or as an open access dataset available on the Open Science Framework (https://osf.io/84tu2/).

## Protocol 2: Generation of castration-sensitive and castration-resistant LNCaP tumor xenografts

This protocol describes the procedure for creating LNCaP prostate cancer xenografts in rodents, and then uses surgical castration to differentiate tumor groups. Upon castration, tumors will either regress (castration-sensitive), or else continue to grow (castration-resistant). Tumor tissue from both intact and castrated mice (including tumors that are both castration-sensitive and -resistant) will be harvested at the end of the experiment for total RNA isolation and subsequent down-stream gene expression analysis in Protocol 3.

### *Sampling*

- The experiment will use at least 5 mice per group, for a power of at least 85.92%.
  - See Power Calculations for details.
  - To buffer against unexpected mouse deaths, 6 mice per cohort will be used.
- The experiment consists of two cohorts:
  - Cohort 1: non-castrated tumor-bearing mice
  - Cohort 2: castrated tumor-bearing mice
    - ∗ 70% of castrated mice develop castration-resistant tumors (C Massie, pers. comm., 2015). Thus, to generate at least 6 castration-sensitive tumors (or 30% of total castrated mice) and at least 6 castration-resistant tumors, 22 mice will be castrated.
  - Total mice:
    - ∗ 28 mice injected with LNCaP cells
    - ∗ 22 injected mice will be surgically castrated

### Materials and reagents

| Reagent | Type | Manufacturer | Catalog # | Comments |
|---------|------|--------------|-----------|----------|
| HC-matrigel | Reagent | Corning | 354262 | Original catalogue number from BD unspecified |
| LNCaP cells, clone FGC | Cells | ATCC | CRL-1740 | Original unspecified |
| NOD/SCID-gamma (NSG) mice | Mice | Jackson Labs | | Original unspecified |
| 1 mL insulin syringe with attached needle; 29G × 1/2 in. | Labware | BD Biosciences | 329411 | Original brand not specified |
| Trypsin | Reagent | Sigma-Aldrich | T6567 | Original unspecified |
| PBS | Reagent | Life Technologies | 14190 | Original unspecified |
| FBS | Reagent | Sigma-Aldrich | R0908 | Original unspecified |
| RPMI-1640 | Synthetic androgen | ATCC | 30-2001 | |
| Ketamine | Drug | Will be left to the discretion of the replicating lab | | Original unspecified |
| Xylazine | Drug | Will be left to the discretion of the replicating lab | | Original unspecified |
| Isoflurane | Drug | Will be left to the discretion of the replicating lab | | Original unspecified |
| Buprenorphin | Drug | Will be left to the discretion of the replicating lab | | Original unspecified |
| Topical antibiotic | Drug | Will be left to the discretion of the replicating lab | | Original unspecified |
| Isopropanol | Reagent | Sigma-Aldrich | W292907 | Original unspecified |
| TRIzol® Reagent | Reagent | Life Technologies | 15596 | Original not specified |
| Silicon-carbide beads | Equipment | BioSpec | 11079110sc | Original unspecified |
| Retsch Mixer Mill | Equipment | Retsch | MM400 | Original unspecified |

### Procedure

Notes:

- All cells will be sent for mycoplasma testing and STR profiling

1. Culture LNCaP cells in RPMI-40 supplemented with 10% FBS at 37 °C/5% $CO_2$.
   (a) Trypsinize to dissociate cells
   (b) Centrifuge at $\leq 1,000$ rpm and remove the supernatant
   (c) Use a P200 pipette to gently dissociate the cells. Pipette up and down several times.
   (d) Count cells
   (e) Resuspend $2 \times 10^6$ dissociated cells in 0.1 ml cold PBS, then mix with an equal volume of cold Matrigel. Total volume should be 0.2 ml per injection.
      i. Approximately $6 \times 10^7$ cells will be needed for all 28 injections

2. Subcutaneously inject mice in the rear flank. Each mouse should receive a single injection.
   (a) Mice should be microchipped prior to injection, so that they can be easily monitored throughout the duration of the study.
   (b) Inject 0.2 ml cell/matrigel mixture per mouse subcutaneously into the flank of the mice using a 29G insulin syringe.

3. Measure tumor volume with manual calipers [#]twice weekly.
   (a) Note time for tumors to form as well as tumor diameter and volume.
      i. [#]Measurement 1 will be width, measurement 2 will be the length, and measurement 3 will be the height of the tumor.
      ii. [#]Since the growth of the tumor is not uniform, we will use the formula $V = length \times width \times height \times 0.5326$ to obtain tumor volume.

4. Let tumors grow to approximately 100 mm$^3$. Tumor growth takes approximately 4 weeks.

5. Randomly assign mice to either the intact or castrate cohort (ratio $\sim$1:5).
   (a) [#]At the time of randomization (when the tumor growth in majority of mice is 100 mm$^3$), mice without any measurable tumors and those with tumor volume between approximately 400–500 mm$^3$ will be eliminated.
   (b) [#]The remaining mice will be arranged in ascending order based on the tumor volume and group numbers will be assigned in a serpentine order in such way that the average tumor volume in each group will have equal or similar tumor volumes.

6. For the castration cohort, surgically castrate mice.
   (a) [#]Castration procedure:
      i. Anesthetize animal with 100 mg/kg ketamine/5 mg/kg Xylazine
      ii. Place on dorsal side facing the tail toward the surgeon. Prep the abdominal area by shaving the hair and swabbing with the surgical swab.
      iii. An abdominal incision will be made and vas deferens and spermatic blood vessel is exteriorized and cauterized and testis tissue removed.

      iv. The tissue is replaced into abdominal cavity and the incision will be closed with wound clips or absorbable sutures.

      v. After surgery, the mice will be given an analgesic, Buprenorphin 0.1 mg/kg (IP or IM) and a topical antibiotic mixture containing bacitracin zinc, neomycin sulfate, and polymyxin b sulfate.

      vi. Castrated animals will be monitored daily for 7 days post castration.

  (b) Around 70% of castrated mice will develop castration resistant tumors, which, after an initial regression, will regrow.

7. Continue monitoring tumor volume in all cohorts #twice weekly until tumor reaches 10% of body weight.

8. Once tumor volume is ≥10% of body weight, sacrifice mouse and excise tumor.
  (a) Euthanize mice under isoflurane anesthesia.
  (b) Spray tumor-bearing area on the flank with 70% isopropanol.
  (c) Make a small incision on the skin of the flank, and peel skin to expose the subcutaneous tumor.
  (d) Excise tumor using surgical scissors and forceps.
  (e) Excess non-tumor tissue will be cleaned from tumor and the tumor will be flash frozen in liquid nitrogen and stored in −80 °C until further processing.

9. Dissociate tissue #with bead homogenizer with 1.00 mm silicon-carbide beads in TRIzol and store homogenized tissue at −80 °C until RNA extraction in Protocol 3.

### *Deliverables*

- Data to be collected:
  - All mouse health records, including date of injection, date of castration, date of sacrifice, reason for sacrifice
  - All tumor volume measurements for all mice
- Samples delivered for further analysis:
  - #Homogenized tissue in TRIzol for use in Protocol 3, stored at −80 °C

### *Confirmatory analysis plan*

- See Protocol 3 for analysis of gene expression

### *Known differences from the original study*

All known differences in reagents and supplies are listed in the materials and reagents section above, with the originally used item listed in the comments section. Tumor volume will be measured twice weekly instead of daily as suggested by the replicating lab. All differences have the same capabilities as the original and are not predicted to alter experimental outcome.

### *Provisions for quality control*

All data obtained from the experiment—raw data, data analysis, control data and quality control data—will be made publicly available, either in the published manuscript or as an open access dataset available on the Open Science Framework (https://osf.io/84tu2/).

## Protocol 3: Assessing expression of the core 16 gene-set by qRT-PCR

This protocol describes how to measure the expression levels of the core 16-gene set using semi-quantitative RT-PCR. This technique will be used to assess gene expression from samples generated in Protocols 1 and 2.

### Sampling

- This experiment consists of five cohorts (2 cohorts from Protocol 1 and 3 cohorts from Protocol 2):
  - Cohort 1: cDNA from untreated LNCaP cells at 0 and 24 h [Protocol 1]
    - $n = 3$ at each time point
  - Cohort 2: cDNA from LNCaP cells treated with R1881 at 0 and 24 h [Protocol 1]
    - $n = 3$ at each time point
  - Cohort 3: cDNA from tumors in intact mice [Protocol 2]
    - $n = 6$
  - Cohort 4: cDNA from castration-sensitive tumors in castrated mice [Protocol 2]
    - $n = \text{TBD}$
  - Cohort 5: cDNA from castration-resistant tumors in castrated mice [Protocol 2]
    - $n = \text{TBD}$

### Materials and reagents

| Reagent | Type | Manufacturer | Catalog # | Comments |
|---|---|---|---|---|
| RNeasy Kit | Reagent | Qiagen | 74106 | Original source unspecified |
| qScript cDNA synthesis kit | Reagent | Quanta Biosciences | 95047 | Original not specified |
| SYBR green mastermix | Reagent | Life Technologies | 4472908 | |
| Oligos for qRT-PCR | Primers | Synthesis left to the discretion of the replicating lab and will be recorded later | | |
| StepOnePlus real-time PCR system | Equipment | Applied biosystems | | |
| 384-well PCR Plates | Labware | Specific brand information will be left up to the discretion of the replicating lab and recorded later | | |

### *Procedure*

1. [#]Extract total RNA from TRIzol-preserved samples using RNeasy kit according to manufacturer's instructions.
   (a) Report total concentration and purity of isolated total RNA.
2. Reverse transcribe [#]0.5 ug RNA from cell and tissue samples derived from Protocols 1 and 2 into cDNA using the [#]qScript cDNA synthesis kit according to the manufacturer's protocol.
3. Perform qPCR analysis of target genes in 384-well plates. *Run each reaction in triplicate.
   (a) Use SYBR green mastermix with the following primer sets according to manufacturer's protocol:
       i. *PEC1*:

    A. F: GCCGGTTGAACATGATCTTT
    B. R: ATGGGCTGAGGTTGTTTGTC
  ii. *TNFSF10*
    A. F: TGTGTCAGGGCTCTACTGTGA
    B. R: ATTCCCAGGGTAGGAGGAGA
  iii. *ABHD12*
    A. F: TAGCCCAGGCGTGTAATAGG
    B. R: CTGGCCTTGAAGCAACATCT
  iv. *XRCC3*
    A. F: GCAATCACAGCCAGAACAGA
    B. R: CAGAAGCAGAGTGTCCCACA
  v. *MAD1L1*
    A. F: GCACCCCTGTTGTTTTCATT
    B. R: ATGCCTGTTCCTTTGTGACC
  vi. *SEC61A1*
    A. Replicating lab will design and optimize primers for this gene
  vii. *GFM1*
    A. F: AAAAGGACTCCCTTCCCTCA
    B. R: ACGCGAGGAAAAAGAGAGTG
  viii. *TSPAN13*
    A. Replicating lab will design and optimize primers for this gene
  ix. *STIL*
    A. F: TCGACCAATCCCAAGTCTTC
    B. R: ATAGAGCACTTCCGGCTTCA
  x. *TRMT12*
    A. F: CTCAAGCAAGCGCATCAATA
    B. R: GGGCTTCCCACTTTCTCTCT
  xi. *EIF2B5*
    A. F: CAGACAGATCGGGTTCCAAT
    B. R: TTCCATTGAGCGCTGATTTT
  xii. *TM4SF1*
    A. F: TGCATTCATTTGGATTGGAA
    B. R: GAAAATCCGACAACCTGGAA
  xiii. *NDUFB11*
    A. Replicating lab will design and optimize primers for this gene
  xiv. *SLC26A2*
    A. F: GGAAAGGGAAGGAAAGGAAG
    B. R: TAGCCACAGCCAGTCACATC
  xv. *AGR2*

       A. F: CAGCCATTCAAATCCCTTGT

       B. R: AAGAGTTCGTGGGGAAATCA

   xvi. *NT5DC3*

       A. F: ATGCACATCTTGGGAAGGTC

       B. R: TCCCTCCCCTTTTCCTCTTA

   xvii. CAMKK2 (known AR target as additional control; *Massie et al., 2011*)

       A. F: TGAAGACCAGGCCCGTTTCTACTT

       B. R: TGGAAGGTTTGATGTCACGGTGGA

   xviii. GAPDH control primers (Wang et al., 2012)

       A. F: GAAGGTGAAGGTCGGAGTC

       B. R: GAAGATGGTGATGGGATTTC

(b) Reaction mixture:

   i. Set up according to the manufacturer's protocols for SYBR Green mastermix

(c) Run qPCR reactions on an [#]ABI StepOnePlus qPCR machine with the following cycling parameters:

   i. 2 min at 50 °C

   ii. 10 min at 95 °C

   iii. 40 cycles of:

       A. 15 s at 95 °C

       B. 1 min at 60 °C

(d) Analyze qPCR data using the $\Delta$Ct method normalized using the Actin control.

### *Deliverables*

- Data to be collected:
  - Quality control data for total RNA and synthesized cDNA
    - $A_{280}/A_{260}$ and $A_{260}/A_{230}$ ratios for both
  - Efficiency calculations for each primer pair
  - Melt curve analysis and optimization data for primer pairs
  - Raw and normalized (to control gene) qRT-PCR values
  - Data analyzed with the $\Delta\Delta$ Ct method

### *Confirmatory analysis plan*

This replication attempt will perform the statistical analyses listed below, compute the effects sizes, compare them against the reported effect size in the original paper and use a meta-analytic approach to combine the original and replication effects, which will be presented as a Forest plot.

- Statistical Analysis of the Replication Data:
  - Comparison of average gene expression for the core 16 genes in xenografts from full, castration-sensitive and castration-resistant tumors
    - One-way ANOVA followed by planned pairwise comparisons using the Bonferroni correction to account for multiple comparisons:

○ Full vs. castration-sensitive

○ Castration-sensitive vs. castration-resistant

- Meta-analysis of original and replication attempt effect sizes:
  ○ This replication attempt will perform the statistical analysis listed above, compute the effects sizes, compare them against the reported effect size in the original paper and use a meta-analytic approach to combine the original and replication effects, which will be presented as a forest plot.
- Additional Exploratory Analysis of the Replication Data:
  ○ MANOVA with the average gene expression for 2 groups of genes in cultured cells as dependent variables and R1881 exposure as the independent variable followed by the following Bonferroni corrected comparisons:
    * Cohort 1: genes with no in vitro regulation by androgens (EIF2B5, NDUFB11, TNFSF10, XRCC3, TRMT12, STIL, AGR2, ABHD12, TSPAN13, MAD1L1) in R1881 treated vs. controls
    * Cohort 2: genes that showed up-regulation by androgens (SLC26A2, GFM1, SEC61A1, TM4SF1, NT5DC3, PECI) in R1881 treated vs. controls
  ○ Comparison of relative gene expression averaged within each sample (in xenografts and in cultured cells) for the core 16 genes: Two-way ANOVA with Cell Type (Xenograft vs Culture) and Treatment (Androgen exposure/Full or Control/Castration-sensitive as main effects followed by planned pairwise comparisons using the Bonferroni correction to account for the following comparisons:
    * Control LNCaP cells vs. R1881 treated LNCaP cells
    * Full xenograft vs. R1881 treated LNCaP cells
    * Castration-sensitive xenograft vs. Control LNCaP cells

### Known differences from the original study

The original study used multiple methods for gene expression analysis, including Illumina expression arrays and qRT-PCR. This replication will only use qRT-PCR to analyze and compare gene expression of the core 16-gene set. This study will include a positive control by measuring expression of a known androgen target (CAMKK2). All known differences in reagents and supplies are listed in the materials and reagents section above, with the originally used item listed in the comments section. All differences have the same capabilities as the original and are not predicted to alter experimental outcome.

### Provisions for quality control

All data obtained from the experiment—raw data, data analysis, control data and quality control data—will be made publicly available, either in the published manuscript or as an open access dataset available on the Open Science Framework (https://osf.io/84tu2/).

## POWER CALCULATIONS

Details of the power calculations are available on the Open Science Framework (https://osf.io/9xhqs/?view_only=fac583dc5ea3471c88d581f64feda258).

## Protocol 1

No power calculation necessary.

## Protocol 2

### Summary of original data

- Note: data were estimated from published heat map.
  - A greyscale intensity value was generated for each gene in each sample using Image J. That value was normalized with respect to white (set to 0) and color (positive numbers for red, negative numbers for blue).
  - These scores were averaged across all 16 genes to generate a total score for each sample.
  - The samples in each group were averaged and a mean and SD generated.
  - See the Data Estimation worksheet (https://osf.io/9xhqs/?view_only=fac583dc5ea3471c88d581f64feda258) for details.

| Figure 7C | Mean score | SD | N |
|---|---|---|---|
| Full | 0.63 | 0.10 | 4 |
| Castration-sensitive | −0.79 | 0.56 | 4 |
| Castration-resistant | 0.32 | 0.44 | 4 |

### Test family

- One-way ANOVA on the mean scores per group followed by Bonferroni corrected comparisons on the following tumor groups:
  - Full vs. castration-sensitive
  - Castration-sensitive vs. castration-resistant

### Power calculations

- Power calculations were performed using GraphPad PRISM v6 for Mac and G*Power v. 3.1.7 (*Faul et al., 2007*).

**One-way ANOVA.**

| F(2,10) | Partial eta$^2$ | $\alpha$ | Effect size $f$ | Power | Total samples | N per group |
|---|---|---|---|---|---|---|
| 12.86 | 0.740783 | 0.05 | 1.690495 | 93.81% | 9 | 3[*] |

Notes.
[*] With 5 samples per group, based on planned comparisons, achieved power is 99.96%.

**Bonferroni corrected $t$-tests.**

| Group 1 | Group 2 | Effect size $d$ | $\alpha$ | Power | N per group |
|---|---|---|---|---|---|
| Full | Castration-sensitive | 3.51191 | 0.025 | 88.80% | 3[*] |
| Castration-sensitive | Castration-resistant | 2.19747 | 0.025 | 85.92% | 5 |

Notes.
[*] With 5 samples per group, achieved power is 99.78%.

### Protocol 3

No power calculation necessary.

## ACKNOWLEDGEMENTS

The PMFRI core team would like to thank the original authors, in particular Dr. Charlie Massie and Dr. David Neal, for generously sharing critical information as well as reagents to ensure the fidelity and quality of this replication attempt.

### Funding

The PCF Movember Foundation Reproducibility Initiative is funded by the Prostate Cancer Foundation and the Movember Foundation. The funders had no role in study design, data collection and analysis, decision to publish, or preparation of the manuscript.

### Grant Disclosures

The following grant information was disclosed by the authors:
Prostate Cancer Foundation and the Movember Foundation.

### Competing Interests

Elizabeth Iorns, Fraser Tan, Joelle Lomax and Nicole Perfito are employed by and hold shares in Science Exchange Inc. The experiments presented in this manuscript will be conducted by Denise Chronscinski at Noble Life Sciences, Inc., which is a Science Exchange lab. No other authors disclose conflicts of interest related to this manuscript.

### Author Contributions

- Denise Chronscinski wrote the paper, prepared figures and/or tables, reviewed drafts of the paper.
- Srujana Cherukeri wrote the paper, reviewed drafts of the paper.
- Fraser Tan, Nicole Perfito and Joelle Lomax conceived and designed the experiments, analyzed the data, contributed reagents/materials/analysis tools, wrote the paper, reviewed drafts of the paper.
- Elizabeth Iorns conceived and designed the experiments.

### Data Availability

The following information was supplied regarding the availability of related data:
Open Science Framework:
https://osf.io/ih9qt/?view_only=c0adeb5f3b6f41ae89fb544725ad97f3.

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
