# Peer review of "Registered report: the androgen receptor induces a distinct transcriptional program in castration-resistant prostate cancer in man"

_PeerJ, doi:10.7717/peerj.1231_

## Round 0.1 · original submission · Minor Revisions

As you will see, both reviewers have similar suggestions concerning the details of your protocols. Please address their comments in the revisions of the protocols. Furthermore, please expand the Introduction in order to discuss the choice of experiments to be replicated.

Reviewer 1 ·

Basic reporting

NO Comments

Experimental design

NO Comments

Validity of the findings

NO Comments

Additional comments

There are two main findings from the Cancer Cell paper by Sharma et al. (2014). First, they identified an androgen receptor target gene signature of 16 genes that predicts castrate-resistant prostate cancer in clinical samples. Second, this gene signature is only present in samples in vivo, but not in cells cultured in vitro. This is of particular relevance, because many of the studies and mechanism described for this disease have been carried out in cultured cells.

The first finding is addressed by Chroscinski and colleagues in the literature summary. The author’s experimental design is thorough and well-articulated. These experiments include xenografts using LNCaP cells, similar to what was reported by Sharma et al., and are designed to test the reproducibility of their findings. The authors address the second finding by including an analysis of cultured LNCaP cells treated with the androgen receptor agonist R1881. Although this experiment was not part of the original report by Sharma et al it will test the validity of their conclusions.

The Experimental Design is robust and with statistical significance to replicate what is reported in Figure 7C of Sharma et al. However, some questions arise:

1.- Given that Sharma et al. did not specify most of the materials used in their published work, the authors should describe how they determined that their procedures will be comparable. The authors state in the Acknowledgements that some critical information and reagents were provided by Sharma and collaborators. The authors should indicate precisely which information and reagents these were.
2. Protocol 1: The authors should explain why they chose to treat the LNCaP cells with R1881 for 3 days, when other reports used shorter time points for the same experiments including Sharma et al (Fig 7G) and Massie et al (2011). A 12-hour time point with CAMKK2 (Massie et al., 2011) as positive control should be included in this analysis, or used instead of the 3 days. This way, Fig.7G would in essence also be replicated by this Reproducibility Study,
3.- Protocol 2: Authors should provide information (such as a citation) on how they know that 70% of the xenografts will develop castration-resistant tumors and what is the expected time till relapse.
4.- Protocol 3: The Deliverables should include information on how the specificity of the primer sets was determined.
5. Protocol 3: Known AR target genes should be included as control to assess the response of cultured LNCaP cells to R188. Some known AR responsive genes such as PSA or FN1 could be found in Wang et al (PMID:22315407).
6.- Protocol 1: It is recommended that the RPMI Medium for treatment of LNCaP cells will be phenol-red free as used in other similar publications.
7.- Protocol 2: Are the castrated mice going to be treated with topical antibiotics or analgesics after surgery?

·

Basic reporting

All comments included in the "General Comments for the Author" section, as suggested by PeerJ staff via email.

Experimental design

All comments included in the "General Comments for the Author" section, as suggested by PeerJ staff via email.

Validity of the findings

All comments included in the "General Comments for the Author" section, as suggested by PeerJ staff via email.

Additional comments

This is an important replication of an in vivo study of AR binding in CRPC tissue and will provide data on the reproducibility of some aspects of the original work. There are some aspects of the design that need to be adjusted to ensure success and some addition controls that should be included (detailed below).

The proposed study aims to examine the reproducibility of AR-regulated expression in 2D culture versus xenografts, however it will not address some of the other key findings of the paper. Notably additional replication studies are needed to: 1/ expand the AR binding profile of human PC/CRPC tumour samples or in xenografts (using ChIP); 2/ dissect the in vivo signals that underpin AR reprogramming ; 3/ assess the clinical utility (and the functional importance) of the 16-gene signature and other genes identified as AR-regulated in CRPC. These may be beyond the scope of this initial replication study, but these should be considered and at least discussed in the subsequent manuscript.

General comments
The original study used chromatin immunoprecipitation of prostate tissue samples from men with prostate cancer and used gene expression data from cell lines, xenografts and clinical samples to define subsets of targets for subsequent bioinformatics analysis (e.g. motif enrichment, functional annotation, clustering analysis). Important conclusions from the original study include the identification of in vivo AR-regulated genes in CRPC tissue, a subset of which were not found to be AR-regulated in vitro. Compared to in vitro derived AR target genes these in vivo derived AR-targets showed divergent functional annotations, co-enrichment of transcription factor binding and expression in CRPC tissue samples. In addition a 16-gene signature was proposed to predict CRPC and recurrent PC better than in vitro derived AR gene signatures. This was suggested to reflect ongoing AR signaling in CRPC that would otherwise be under-estimated by measuring the expression of in vitro derived AR gene sets.

The authors of this Reproducibility Initiative study have chosen to focus on the 16-gene signature presented in the manuscript, in particular comparing the in vitro (2D-culture) and in vivo (xenograft) androgen regulation of this gene signature. The core 16-gene signature from the original study was derived by intersecting AR binding in CRPC tissue, with genes down-regulated in castrated xenografts and up-regulated in CRPC tissue samples. It is important to note that this signature was not selected to discriminate in vitro AR-regulated from in vivo AR-regulated genes, indeed this gene signature showed gene set enrichment for in vitro AR-regulated genes (original paper Figure 7E). From the original study:

“Selection of the genes with the most consistent changes across castrated xenografts and CRPC identified a core 16 gene signature (Table S4).”

“Gene-set analysis and direct comparison of gene-expression changes showed that this core 16 gene set had high gene-set enrichment scores for both in vitro AR-regulated genes and xenograft castration-regulated genes, similar to a published 250 gene ‘‘AR activity signature’’ (Mendiratta et al., 2009) (Figure 7E), but overall showed a stronger correlation with AR regulation in vivo (Figures 7E–7H; Figure S2).”

However, consistent with the overall conclusions of the original study, a large proportion of the genes in this 16 gene signature (10/16, detailed below) do not show evidence of androgen-regulation in 2D culture (from our LNCaP gene expression data). Therefore, if this is accounted for the study will still provide an important replication of the core findings of the original study (i.e. that there exists a distinct set of AR-regulated genes in vitro and in xenografts, reflecting a dynamic, context-dependent activity of the AR).

In summary, this Reproducibility Initiative study will provide insights into the reproducibility of one aspect of the original work (i.e. at the transcriptional level for a core set of genes). This work should provide some direction for future efforts to study the activity and in vivo AR reprogramming, the underlying signals that drive this and the utility of the in vivo targets identified.

Specific comments relating to the suggested review format:
● Do the experiments chosen embody the main conclusions drawn from the original article?

The proposed study will address some of the main conclusions of the original study and will hopefully prompt future studies that will probe further the in vivo signalling of the AR in CRPC tumour tissue.

● Do the authors accurately summarize the literature with respect to other direct replications of the experiments being replicated?

For the most part the authors accurately summarize the literature regarding direct replications of the original study. However, relating to the comment in the introduction on lines 75-76 it should be noted that the regulation of genes in the 16-gene signature was shown in the original study using two distinct PC xenografts run in different labs (i.e. LNCaP xenografts by Sharma, et al 2013 and KUCaP xenografts from a re-analysis of the data originally presented by Terada, et al 2010). Therefore, some degree of replication has already been undertaken with regards to transcript regulation in xenografts.

Also in the introduction on lines 82-85 the authors state:
“While no direct replications exist in the literature for genome-wide profiling of AR binding sites, parts of Sharma and colleagues’ work have been reproduced in other studies. Tao and colleagues profiled AR target genes in cultured LNCaP cells but reported gene ontology only (Tao et al. 2014).”

It should be clarified in the introduction that a large number of studies have carried out genome-wide profiling of AR binding, notably the study from which Tao, et al obtained data for their re-analysis. The following studies should be cited here:
Yu, et al. Cancer Cell 2010;17(5):443-54. PMID: 20478527
Wang Q, et al. Cell. 2009; 138(2):245-56. PMID: 19632176
Bolton EC, et al. Genes Dev. 2007; 21(16): 2005-2017. PMID:17699749
Jia L, et al. PLoS One. 2008;3(11):e3645. PMID:18997859

● Are the proposed experiments appropriately designed?

The experiments are well designed in most aspects, however there are some amendments that should be made to ensure success of the experiments.

In Protocol 1 LNCaP cells should be expanded in RPMI supplemented with 10% FBS as described, but once cells are at ~70% confluence media should be removed, cells washed twice with 1x PBS and replaced with RPMI media supplemented with 10% charcoal dextran treated FBS. Cell should be kept in this steroid depleted media for 72h prior to androgen stimulation or vehicle control treatment. This provides a treatment contrast for AR activation studies (i.e. subsequent gene expression studies in this case).

In addition we and others have used much shorter androgen stimulation times than the 72h suggested in Protocol 1 (e.g. AR binding can be profiled at 4h and gene expression at 24h). It could be informative to have a short time-course for expression only studies such as the one proposed here (e.g. 0h, 8h, 16h, and 24h following treatment). At later time-points (>24h) it becomes impossible to distinguish direct and indirect expression changes following a given stimulus.

Finally, the selection of the FGC clone of the LNCaP cell line is appropriate for this reproducibility study, however it should be noted that the original study did not use this FGC clone but rather low passage LNCaP cells. However, it is unclear whether this will affect the outcome of the proposed studies.

● Are the proposed statistical analyses rigorous and appropriate?

The analyses proposed seem appropriate.

● What can the replication team do to maximize the quality of the replication?

As explained above the 16-gene signature was not selected to discriminate in vitro from in vivo regulated AR target genes (rather they were selected to identify AR bound genes in CRPC tissue that are up-regulated in CRPC tissue and down-regulated following castration in xenografts). Therefore to address their specific question regarding in vivo regulated AR targets the authors should group genes in this signature by their AR regulation in vitro (i.e. when comparing expression in 2D-cultured and xenografted LNCaPs). Of the 16 genes in this signature 10 show no evidence of in vitro regulation by androgens in LNCaP cells (EIF2B5, NDUFB11, TNFSF10, XRCC3, TRMT12, STIL, AGR2, ABHD12, TSPAN13, MAD1L1), three show marginal up-regulation after 24h treatment (SLC26A2, GFM1, SEC61A1) and three show moderate-high up-regulation by 24h treatment (TM4SF1, NT5DC3, PECI). Alternatively the authors could select a larger set of genes that showed AR binding in CRPC tissue, AR-regulation in castrated xenografts but were not androgen regulated in 2D-cultured LNCaP cells (i.e. from the original study Figure 3, right hand panel).

Further controls should also be included to assess the efficacy of androgen stimulation and withdrawal in each experiment. For example qrtPCR should be carried out for established AR-regulated genes to assess the activation of AR signalling in 2D-cultured cells (e.g. one or more of PSA/KLK3, KLK2, FKBP5 and TMPRSS2).

As suggested above a time-course (even taking only 2-3 time-points) would increase the confidence in the data generated and could help to capture or rule-out dynamic changes that would be missed by a single time-point.

The authors could also explore the possibility of qrtPCR analysis of the gene signature and control genes using tissue from human CRPC tumours to assess the utility of the signature for its proposed application.

Finally, (dependant on funds and/or available collaborators) the authors could consider ChIP-qPCR or ChIP-sequencing using the valuable xenograft material that they will generate as part of the proposed study. This could further examine the re-programming of AR binding in xenografted LNCaP cells compared to the wealth of data available for LNCaP cells grown in 2D culture.

---

## Round 0.2 · accepted · Accept

Please proceed with the experiments as outlined in this Registered Report. We look forward to receiving the resulting manuscript for this Replication Study.

Reviewer 1 ·

Basic reporting

No comments

Experimental design

No comments

Validity of the findings

No comments

Additional comments

Chroscinski and colleagues have addressed the questions raised in previous revision and now the experiments proposed in their manuscript will allow testing the reproducibility to what was reported by Sharma et al. Thus, the manuscript by Chroscinski et al. should be accepted without further revisions.

·

Basic reporting

The authors have addressed adequately all of my comments from the first review.

Experimental design

The authors have addressed adequately all of my comments from the first review.

Validity of the findings

The authors have addressed adequately all of my comments from the first review.

Additional comments

The authors have addressed adequately all of my comments from the first review.